# Brief Communication: On the influence of vertical wind shear on the combined power output of two model wind turbines in yaw

Jannik Schottler, Agnieszka Hölling, Joachim Peinke, and Michael Hölling

ForWind, University of Oldenburg, Institute of Physics, Oldenburg, Germany

*Correspondence to:* Jannik Schottler (jannik.schottler@forwind.de)

**Abstract.** The effect of vertical wind shear on the total power output of two aligned model wind turbines as a function of yaw misalignment of the upstream turbine is studied experimentally. It is shown that asymmetries of the power output of the downstream turbine and the combined power of both with respect to the upstream turbine's yaw misalignment angle can be linked to the vertical wind shear of the inflow.

## 1 Introduction

Lately, different concepts of active wake control are discussed throughout the research community. One promising concept is the wake deflection by intentional yaw misalignment of single wind turbines. The principle of deflecting the velocity deficit behind a wind turbine was observed in field measurements by Trujillo et al. (2016), in wind tunnel experiments (e.g. Medici and Alfredsson, 2006; Krogstad and Adaramola, 2012) and in numerical simulations (e.g. Jiménez et al., 2010; Gebraad et al., 2014; Vollmer et al., 2016). Further, Gebraad et al. (2014) and Fleming et al. (2016) applied the concept to wind farm control strategies using large-eddy simulation (LES) methods, showing a potential power increase in wind farm applications.

Vollmer et al. (2016) report on an asymmetric deflection of a turbine's wake with respect to its direction of yaw misalignment in numeric studies. Similarly, Bastankhah and Porté-Agel (2016) found that a wake moves upwards or downwards depending on the direction of a yaw misalignment using PIV measurements behind a small turbine model. This observation is explained by an interaction of the wake's rotation and a pair of counter-rotating vortices formed in yawed conditions with the ground.

Vollmer et al. (2016) studied the influence of atmospheric stabilities on the wake deflection by yaw misalignment. The results show that different stratifications indeed result in varying deflections of the wake behind the rotor of a numeric turbine model. More precisely, disparities between wake deflections due to yaw misalignments of $+30°$ and $-30°$ were significantly different considering different atmospheric stratifications and therewith different shears. It is believed that a combination of a vertical inflow gradient, the wake's rotation and the wind veer cause asymmetric wake deflections with respect to the rotor's yaw angle.

Examining the power of a turbine array, Fleming et al. (2014) and Gebraad et al. (2014) showed that only one direction of yaw misalignment resulted in a power increase of a two turbine array, while the exact opposite direction caused a power decrease. This finding was confirmed by Schottler et al. (2016) experimentally using two model wind turbines. As those findings impact the applicability of the concept significantly, reasons for the asymmetry need to be understood.

In this study, we show that a vertical wind shear has a direct effect on the power's asymmetry of two model wind turbines during yaw misalignment.

## 2  Methods

The experiments were performed at a wind tunnel of the University of Oldenburg, with an open test section of $1\,\mathrm{m} \times 0.8\,\mathrm{m} \times 5\,\mathrm{m}$

$[\mathrm{w} \times \mathrm{h} \times \mathrm{l}]$. Two model wind turbines as described by Schottler et al. (2016) were used in streamwise displacement. The turbines were separated by $3D$, with $D = 0.58\,\mathrm{m}$ being the rotor diameter and rotate clockwise when observed from upstream. The upstream turbine is placed on a turning table allowing for yaw misalignment, where a positive yaw angle is a counter-clockwise rotation of the rotor when seen from above. The downstream turbine utilizes a partial load control and therewith adapts to the changing inflow conditions. Power measurements are based on the rotational speed and the torque, being proportional to the

electric current of the generator. Further details about the setup, power measurements and turbine control are described by Schottler et al. (2016). In order to isolate the effect of a vertical wind shear in the inflow, the horizontal axes of an active grid (see Weitemeyer et al. (2013)) at the wind tunnel outlet were set statically to create two different inflow profiles, which were characterized prior to the experiments. 13 hot wire probes were used simultaneously in a vertical line arrangement with a distance of $75\,\mathrm{mm}$ separating two sensors. For both settings of the grid, data were recorded for $120\,\mathrm{s}$ at a sampling frequency of

$2\,\mathrm{kHz}$. The array was installed 1 m downstream from the grid at the position of the upstream turbine's rotor, which was installed after characterizing the inflow. Fig. 1 shows mean wind speeds over the height z, whereas $\mathrm{z} = 0\,\mathrm{m}$ corresponds to the bottom of the wind tunnel outlet. The reproducibility of time averaged velocity profiles for one grid setting has been investigated and confirmed. Further, mean values have been checked for statistical convergence. As of now, we refer to the inflow conditions

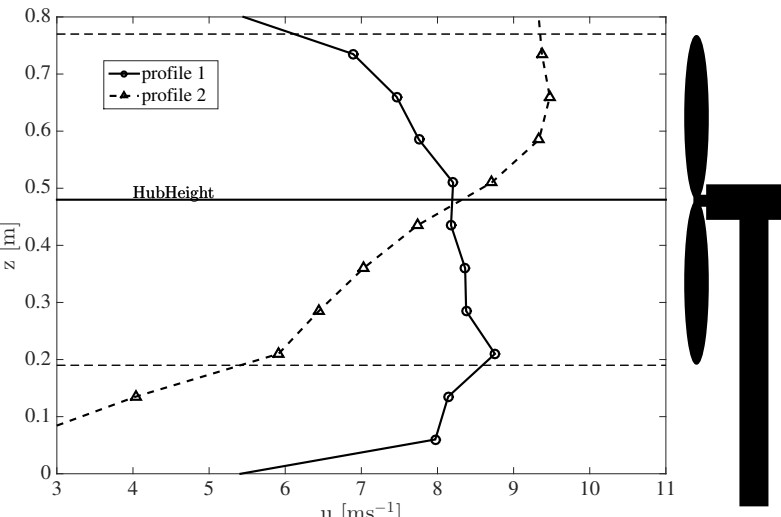

**Figure 1.** Mean velocity values of the vertical wind speed profiles 1 and 2 that were used as inflow conditions. The dashed, vertical lines mark the heights of the rotor tips of the turbine that was installed after characterizing the inflow profiles.

shown in Fig. 1 as *profile 1* and *profile 2*. Using two inflows which feature a vertical wind shear of opposite direction over the rotor area allows for an investigation of the gradient's influence on the asymmetric power output of the two turbines with respect to the upstream turbine's yaw angle, $\gamma_1$.

## 3 Results

Mean values of the upstream turbine's power $P_1$, the downstream turbine's power $P_2$ and their sum $P_{tot}$ are shown as a function of the yaw angle $\gamma_1$ in Fig. 2. Data points are normalized to the respective maximum of $P_{tot}$. Looking at Fig. 2(a), asymmetries

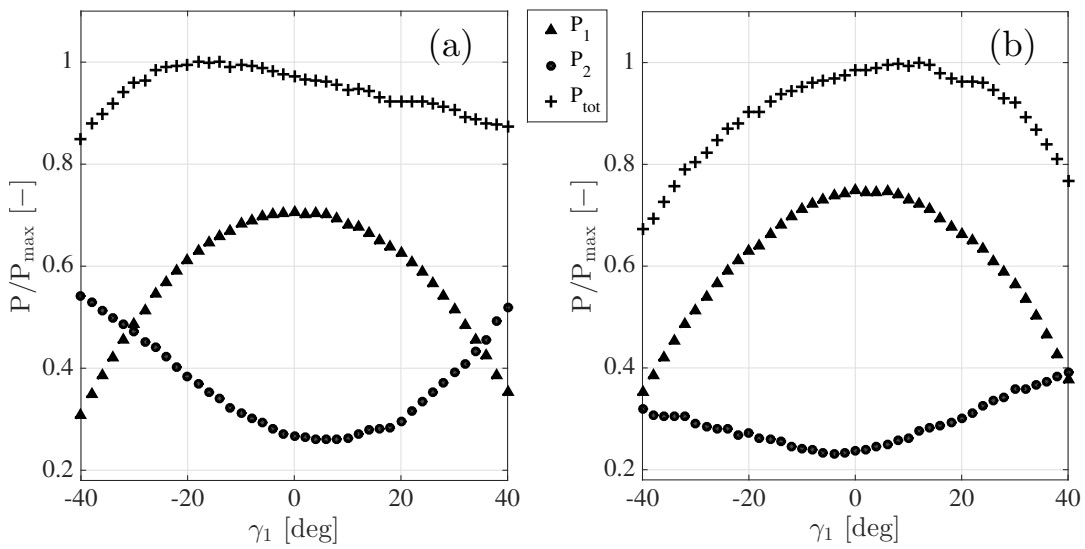

**Figure 2.** Mean values of $P_1$, $P_2$ and $P_{tot}$ for each examined value of $\gamma_1$ during the both inflow condition profile 1 (a) and profile 2 (b).

of $P_2$ and $P_{tot}$ with respect to $\gamma_1$ become obvious. The minimum of the downstream turbine's power $P_2$ is shifted towards positive angles. The maximum of the combined power $P_{tot}$ is at $\gamma_1 \approx -18°$, being approx. 4 % larger compared to the case of no yaw misalignment $\gamma_1 = 0°$. Also the combined power shows a distinct asymmetry with respect to $\gamma_1$. While the power is

maximal at $\gamma_1 \approx -18°$, it further decreases for larger values of $\gamma_1$. For positive yaw angles, the total power output is smaller compared to the case of no yaw misalignment. The results support that the direction of a purposeful yaw misalignment is of great relevance regarding the application of this concept to wind farm control. Further, the general shape of the graphs are in good agreement with numeric simulations of full size turbines reported by Gebraad et al. (2014) and Fleming et al. (2014).

Fig. 2(b) shows the results of the same experiment, whereas nothing but the inflow conditions was changed to profile 2. Since

the reproducibility of results was proven by Schottler et al. (2016), the effect of the changed inflow is isolated. As can be seen, asymmetric shapes of $P_2$ and $P_{tot}$ are still observed. More importantly, the direction of the asymmetry changed with the direction of the inflow's vertical shear. Now, in Fig. 2(b), the minimum of $P_2$ is located at negative yaw angles, $\gamma_1 \approx -4°$. Moreover, the yaw angle direction at which the combined power is maximum changed, being positive ($\gamma_1 \approx 12°$) for inflow

profile 2. Our results show that the reason for the asymmetric shapes of the graphs in Fig. 2 is related to the inflow's vertical wind shear, which is further discussed in Sec. 4.

## 4  Discussion and conclusion

The vast majority of model wind turbine experiments face a Reynolds number mismatch between the laboratory and full scale case, which is nearly a factor of 170 in this study. However, due to the good agreement of the general shapes of the turbines' normalized powers comparing the present study and Schottler et al. (2016) with simulations of a full scale case (Fleming et al., 2014; Gebraad et al., 2014), the Reynolds number dependence is assumed to be rather insignificant when judging general effects of wake deflection. It should be noted that the LES simulations performed in (Fleming et al., 2014) and (Gebraad et al., 2014) include a wind veer, which was not reproduced experimentally and should be kept in mind when comparing the numerical and experimental studies. Further, due to spatial limitations of the wind tunnel, the profiles shown in Fig. 1 are expected to be not fully developed. Therefore, their downstream development, which was not investigated in this study, might impact the wake deflections. This effect could not be isolated. Next, the inflow profiles vary regarding their turbulence intensity. This is expected to impact the wake recovery (Wu and Porté-Agel, 2012), but not the asymmetries in power reported. It should also be noted that the upstream turbine's tip speed ratio (TSR) is not constant for varying angles $\gamma_1$. As shown by Krogstad and Adaramola (2012), the TSR maximizing the power is subject to change with the yaw angle. Therefore, the load control utilized by the downstream turbine was not used for the upstream turbine, which was operated at constant electrical load for both profiles. However, as the upstream turbine's TSR is symmetric with respect to $\gamma_1$, this is not expected to affect the asymmetries observed in this work.

This study investigates the influence of vertical wind shears on the power output of two aligned model wind turbines. An asymmetry of the power output with respect to the upstream turbine's yaw angle was found in prior experiments on laboratory scale (Schottler et al., 2016) as well as in full scale numeric simulations (Gebraad et al., 2014; Fleming et al., 2014). Only one direction of yaw misalignment resulted in a power increase, whereas the exact opposite direction caused a power decrease of the turbine array. For a potential application of active wake control by intentional yawing, this effect needs to be understood. With the present methods, we investigate the reasons for the asymmetric power output of a two turbine array and isolate the effect of a vertical inflow gradient's orientation. A strong linkage between the asymmetry and the velocity gradient's orientation was found. If the reported asymmetry depends on boundary conditions of the surroundings, which our results suggest, then this drastically impacts the applicability to real world wind farm control scenarios. In this study, the downstream turbine's power is used as indicator. The interesting results regarding the asymmetry and its linkage to the inflow conditions motivate further examinations, such as detailed wake measurements during different inflow gradients and yaw errors. As the yaw angle is a distribution in full scale cases, future works should address this issue and its impact on active wake redirection strategies.

*Acknowledgements.* Parts of this work were funded by the Reiner Lemoine Stiftung (RLS), Germany, which is greatly appreciated. The authors thank Stefan Ivanell for providing the rotor blade design as well as Jan Bartl and Lukas Vollmer for fruitful discussions.

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
