# Peer review of "Brief Communication: On the influence of vertical wind shear on the combined power output of two model wind turbines in yaw"

_Wind Energy Science, 2017_

## Referee Comment (RC1) · Anonymous Referee #1 · 14 Mar 2017

Review of the paper

**"Brief Communication: On the influence of vertical velocity profiles on the combined power output of two model wind turbines in yaw"**

This manuscript presents a wind-tunnel study of the influence of vertical velocity profiles on the power production of two model turbines in a row, while the upwind one operates under yaw. The results show that the variation of the combined power production (and also the power produced by the downwind turbine) as a function of the yaw angle of the upwind turbine $\gamma_1$ is asymmetric in a sense that the results are different for positive and negative yaw angles. Two different incoming boundary-layer profiles were employed to show that this asymmetrical behavior is due to the vertical velocity profile of the incoming flow.

There is an untapped potential to control the yaw angle of turbines for the sake of power-production improvements in wind farms. As a result, studying the performance of yawed turbines as well as turbines located downstream is of great importance for the wind energy community. However, there are some major issues with the contribution of this paper as well as the presentation and discussion of the results that need to be first addressed. I therefore believe that the paper in its current form does not merit publication in "WES: brief communications", which is intended for high-impact research. Lists of my major and minors comments are found below:

**Major comments:**

- One of the main criticisms to the paper is the fact that it suffers from the lack of velocity and thrust measurements. For instance, wake measurements at different yaw angles can provide more insights on the asymmetric behavior observed in the power of the downwind turbine. Even only thrust measurements for the upwind turbine can shed lights on the overall strength of the turbine wake, and consequently the performance of the downwind turbine. However, I do appreciate that the authors are motivated to perform velocity measurements in their future research.
- Apart from the yaw angle, the operating tip-speed ratio is very important as it significantly affects the turbine power. It is not clear in the manuscript if turbines always operate at the optimal tip-speed ratio (i.e., the one at which the turbine power is maximum) or a constant tip-speed ratio is used for all the different yaw angles. In other words, please explain how the effect of yaw angle on power production is isolated from the effect of other parameters such as the operating tip-speed ratio.
- The literature review has to be improved. Some very relevant experimental and numerical studies in the literature (e.g., Jimenez et al. 2010, Howland et al. 2016, Bastankhah and Porte-Agel 2016) are not mentioned in the manuscript. In particular, Bastankhah and Porte-Agel (2016) has recently showed that, in addition to the lateral deflection, the wake of a yawed turbine moves vertically, and the magnitude and the direction of both horizontal and vertical displacements depend on the yaw-angle direction. This can explain why the power of the downwind turbine (or the combined power) depends on the yaw-angle direction of the upwind turbine.

- Please explain why a relatively unrealistic spacing between turbines (3D) is selected. In wind farms, turbine spacing usually falls in the range of 5D to 7D depending on terrain and flow conditions.
- There is no information on how the turbine power is measured. Is it the electrical power? Or the mechanical power extracted by the turbine from the wind?
- Please provide more information about the wind tunnel (e.g., wind-tunnel type, test section size, and blockage ratio).
- I suggest the authors to also test the performance of the turbines under uniform inflow conditions as a *reference case*. This can strengthen the authors' arguments. Moreover, *Profile 2* does not have a good quality. It has a positive slope at lower heights and a fairly negative slope at higher heights. A profile with a clearly negative slope (in contrary to *profile 1*) is more constructive.
- Figure 2: Please add the variation of the power with the yaw angle for the upwind turbine. This helps readers to easier realize how yawing the upwind turbine reduces its own power and increases the power of the downwind one.
- Please define which yaw-angle direction is assumed to be positive in this study. Moreover, please specify in the manuscript the rotational direction of the turbine.

**Minor comments:**

- P3, L2: replace "… for every examined …" with "… as a function of …".
- P3, L5: I think it is better to use "maximum" instead of "maximal" here and in the rest of the manuscript.
- P3, L5: "perfect yaw alignment" is a bit vague. Maybe, it can be replaced with "no yaw misalignment".
- P3, L15: replace "during" with "for".
- P3,L16: remove the comma in "our results suggest,".
- P3, L15: you can replace "Also for the total power output, the sign of the maximum's location …" with "Moreover, the yaw-angle direction at which the combined power is maximum …".
- P4, L9: "than" is supposed to be "then".

**Additional references:**

- Jiménez, Á., Crespo, A., & Migoya, E. (2010). Application of a LES technique to characterize the wake deflection of a wind turbine in yaw. *Wind energy*, 13(6), 559-572.
- Howland, M. F., Bossuyt, J., Martínez-Tossas, L. A., Meyers, J., & Meneveau, C. (2016). Wake structure in actuator disk models of wind turbines in yaw under uniform inflow conditions. *Journal of Renewable and Sustainable Energy*, 8(4), 043301.
- Bastankhah, M., & Porté-Agel, F. (2016). Experimental and theoretical study of wind turbine wakes in yawed conditions. *Journal of Fluid Mechanics*, 806, 506-541.

---

## Referee Comment (RC2) · P. van der Laan (Referee) · 15 Mar 2017

**Review of Brief Communication: On the influence of vertical velocity profiles on the combined power output of two model wind turbines in yaw by J. Schottler, et al.**

Reviewer: M. Paul van der Laan, DTU Wind Energy

March 15, 2017

The article presents wind tunnel measurements of two aligned model wind turbines. The influence of two different inflow profiles on the power output is investigated, as function of the yaw misalignment angle of the upstream wind turbine.

You conclude that a negative wind shear moves the combined maximum power output from negative to positive yaw angles compared to a positive wind shear. I think there is a some information lacking to support this conclusion:

1. Where are the profiles from Figure 1 measured with respect to the wind turbine positions and how do they develop from the first to the second wind turbine and further downstream (without the wind turbines present in the tunnel). My concern is that if the wind profiles are far from equilibrium, it could influence the wake deflection significantly.

2. What is the turbulence intensity and/or how do the turbulence profiles look like that correspond to profile 1 and 2 from Figure 1?

It would strengthen the article if a model or simulation results could confirm your experimental results. I think that a simple actuator disk (AD) model with rotation based on airfoil data in a Reynolds-averaged Navier-Stokes (RANS) code should be able to model it. I could not resist to perform such simulations myself. In Figure 1, simulation results are shown for two different wind shears: a small wind shear of $\Delta U = 1$ m/s and a large wind shear of $\Delta U = 3.5$ m/s (taken over the rotor area). The yaw angle is positive for a counter clockwise rotation seen from above. The inflow profiles represents a neutral atmospheric surface layer with a low and a high roughness length (or a turbulence intensity of 5% and 20% at hub height) and a hub height wind speed of 8 m/s in both cases. I have used the NREL-5MW reference wind turbine. Unfortunately, I could not replicate a clear shift in total power as function of yaw. There are small asymmetries, but not as strong as you have presented in the article. In addition, the maximum power of the combined wind turbines lies around 0 deg for both shears, which means that there is no benefit in yawing the first wind turbine in an aligned configuration, at least in my simulations. I have also done RANS AD simulations where the second AD has a slight lateral offset of $\pm D/8$,

[Figure]

Figure 1: RANS AD simulation results of an aligned setup for a low and a high wind shear.

where $D$ is the rotor diameter. This would correspond to a misaligned wind direction of $\pm\mathrm{atan}(1/8) = \pm7°$, with respect to the downstream wind turbine. The results are shown in Figure 2. Figure 2 indicates that your results could be generated by asymmetries in the experimental setup. Possibly your inflow profiles develop downstream, which could result in flow that is not perfectly aligned with the wind turbine pair (before yawing the first wind turbine).

[Figure]

Figure 2: RANS AD simulation results of two wind turbines with a lateral offset of $\pm D/8$.

Other minor comments:

1. A few references include duplicated links.

2. Page 1, lines 12-14: I am not able to find a discussion on asymmetries of wake deflection in Gebraad et al. (2016).

3. I would call vertical velocity gradient simply wind shear.

4. How is your yaw angle defined?

5. Page 2, lines 14-16: I would add over the rotor area to be more precise: Using two inflows which feature a vertical velocity gradient in opposite direction over the rotor area allows an investigation of ....

---

## Author Comment (AC1) · 29 Apr 2017

Authors' response to Anonymous Referee #1:

We, the authors, are very thankful for the detailed and constructive comments and greatly appreciate the willingness to review our manuscript. Please find our responses below. The original comments are shown in **bold** with the respective answers below. Excerpts of the manuscript are shown in *italic writing*, whereas additions are written in blue and deleted parts in <del>red</del>.

Please note that the format of citations in manuscript excerpts might be changed.

Thank you very much for your efforts,

Jannik Schottler on behalf of all authors

Major comments:

1. One of the main criticism to the paper is the fact that is suffers from the lack of velocity and thrust measurements. For instance, wake measurements at different yaw angles can provide more insights on the asymmetric behavior observed in the power of the downwind turbine. Even only thrust measurements for the upwind turbine can shed lights on the overall strength of the turbine wake, and consequently the performance of the downwind turbine. However, I do appreciate that the authors are motivated to perform velocity measurements in their future research.

Thank you very much for the constructive criticism. We do agree that wake velocity measurements and thrust measurements along with the presented power data would give an overall insight in the scenario as a whole. However, wake velocity measurements were not performed in the scope of this manuscript. The focus of this paper are power measurements of both turbines in relation to the upstream turbine's yaw angle and two inflow profiles. In this brief manuscript, we focus on one main message, which is how both inflow profiles affect the asymmetries in the powers during yaw misalignment differently.

We believe that the *whole picture* of active wake control by yaw misalignment can only be grasped by studying the wake evolutions by means of numerous turbulence parameters along with turbine data such as power and loads for various inflow conditions, both experimentally and numerically. In our opinion it is hardly possible nor desirable to cover all of these aspects in one publication. Instead, we believe that it adds clarity, intelligibility and systematics to literature when focusing on few if not one main message only, especially in the manuscript type "Brief communications".

In our manuscript, the main quantity of interest is the power. The reasons for the shapes of the powers in relation to the yaw angle is believed to be complex and cannot be covered in one publication. Recent works such as Bastankhah and Porté-Agel (2016) [1] or Vollmer at al. (2016) [2] show that solely the wake velocities of deflected wakes due to yaw misalignment comprises a challenging complexity.

In our study, the power and therewith the performance of the downstream turbine is measured directly, thus thrust measurements of the upstream turbine would, in our opinion, not contribute significantly to information regarding the downstream turbine's performance.

2. Apart from the yaw angle, the operational tip-speed ratio is very important as it significantly affects the turbine power. It is not clear in the manuscript if the turbine always operate a the optimal tip-speed ratio (i.e., the one at which the turbine power is maximum) or a constant tip-speed ratio is used for all the different yaw angles. In other words, please explain how the effect of yaw angle on power production is isolated from the effect of the other parameters such as the operating tip-speed ratio.

Thank you for pointing this out, indeed the TSR is affecting the wake of a wind turbine and therewith its deflection. In the present setup, the rotational speed of the model wind turbine(s) is controlled using a field effect transistor (FET) within the electric circuit. By applying an external voltage  $U_{\text{FET}}$  to the FET, the electric current is manipulated and therewith the electric load and the rotational speed are controlled. The concept and the settings during the experiment are described in [3], which is why this information is missing the current manuscript, the reference to the description in [3] is given in p.2, ll. 5-6.

During the experiment, the downstream turbine utilizes the active load control, where a PI-controller controls the load by continuously adapting the voltage UFET. Therewith, the turbine automatically adapts to changing inflow conditions, keeping the TSR of the downstream turbine constant. For the upstream turbine, however, the control voltage UFET was kept constant for each yaw angle  $\gamma_1$  and both inflow profiles. This results in a variation of the TSR with  $\gamma_1$ , which is shown in Fig. 1 of this document. Unfortunately, the TSR is not equal for both profiles

Figure 1. TSR  $\lambda_1$  over the yaw angle  $\gamma_1$ , during constant control voltage  $U_{FET}$  and  $u \approx 8 \text{ms}^{-1}$ .

used. However, both profiles do not show any distinct asymmetries. Herewith it is shown that the asymmetries in the power output, which are the focus of this paper, do not result from the TSR variations.

3. The literature review has to be improved. Some very relevant experimental and numerical studies in the literature (e.g. Jimenet et al. 2010, Howland et al. 2016, Bastankhah and Porte-Agel 2016) are not mentioned in the manuscript. In particular, Bastankhah and Porte-Agel (2016) has recently showed that, in addition to the lateral deflection, the wake of a yawed turbine moves vertically, and the magnitude and the direction of both horizontal and vertical displacements depend on the yaw-angle direction. This can explain why the power of the downwind turbine (or the combined power) depends on the yaw-angle direction of the upwind turbine.

Thank you very much for pointing this out. We fully agree that the mentioned studies, especially Bastankhah and Porté-Agel [1] did some very interesting work on the topic, which should be included in the literature review. Amongst other aspects, it was found that the direction of yaw misalignment results in a upward or downward movement of the examined model turbine wakes. A method based on potential theory was used to show that this asymmetric wake deflection for positive and negative yaw angles result of an interaction between a pair of counter rotating voracities, the ground and the wake rotation. For details, please see chapter 3 in [1]. This finding supports our conclusion, that the asymmetry in power of the downstream (and therewith the total power) turbine with respect to  $\gamma_1$  is the result of the wake rotation interacting with shear. Similar assumptions are stated by Gebraad et al (2014) [4]. There, reasons for an initial wake deflection without yaw misalignment ( $\gamma = 0^{\circ}$ ), are given as shown in the quote in Figure 2 of this document. Similar to Bastankhah and Porté-Agel, a combination of the wake's rotation and the interaction with the ground/wind shear is pointed out.

$$\frac{D_{i}}{D_{i}} \left[ \frac{2\kappa_{d} \left[ x - A_{l} \right]}{D_{i}} + 1 \right]$$

In addition, in the simulations described by Fleming *et al.*,23 it was shown that a small lateral wake deflection occurs when the turbine is not yawed (i.e.,  $\gamma_i = 0$ ). This deflection can be explained by vertical shear in the boundary layer and wake rotation. In reaction to the rotor rotating clockwise, the wake will rotate counterclockwise. As a result, the low-speed flow in the lower part of the boundary layer will be rotated up and to the right, and high-speed flow in the upper part of the boundary layer will be rotated down and to the left. Consequently, the velocity deficit at the right part of the wake (looking downstream) increases, so the wake deflects to the right. Because in *SOWFA Simulation Series 1* and 2, the wake behavior was tested for a single mean wind velocity with a limited velocity variation caused by turbulence, the exact dependence of the wake deflection on the rotor speed could not be derived from the power data obtained. Therefore, this rotation-induced wake lateral offset is parameterized through a simple linear function of the downstream distance from the rotor:

Figure 2. Screenshot taken from [4].

Jimenéz et al. did important work on the topic of wake deflection by yawing in general. However, only one direction of yaw misalignment was studied in the mentioned paper and asymmetries are therefore not reported. Nevertheless, this important piece of work should be mentioned in the manuscript.

We suggest to add this works to the literature review as done below:

Lately, different concepts of active wake control are discussed throughout the research community. One promising concept is the wake deflection by intentional yaw misalignment of single wind turbines. The principle of deflecting the velocity deficit behind a wind turbine was observed in field measurements by [5], in wind tunnel experiments [6, 7] and in numerical simulations [4, 2][8, 4, 2]. Further, [9] and [10] applied the concept to wind farm control strategies using large-eddy simulation (LES) methods, showing a potential power increase in wind farm applications.

[2] and [4] report on an asymmetric deflection of a turbine's wake with respect to its direction of yaw misalignment . [11] and [9] showed that only one direction of yaw misalignment resulted in a power increase of a two turbine array, while the exact opposite direction caused a power decrease. This finding has been confirmed by [3] experimentally using two model wind turbines. As those findings impact the applicability of the concept significantly, reasons for the asymmetry need to be understood. in numeric studies. Similarly, [1] found that a wake moves upwards or downwards depending on the direction of a yaw misalignment using PIV measurements behind a small turbine model. This observation is explained by an interaction of the wake's rotation and a pair of counter-rotating vorticies formed in yawed conditions with the ground. [2] studied the influence of atmospheric stabilities on the wake deflection by yaw misalignment. The results showed that different stratifications indeed resulted in varying deflections of the wake behind the rotor of a numeric turbine model. More precisely, disparities between wake deflections due to yaw misalignments of  $+30^{\circ}$  and  $-30^{\circ}$  were significantly different considering different atmospheric stratifications and therewith different vertical velocity gradients. It is believed that a combination of a vertical inflow gradient, the wake's rotation and the wind veer cause asymmetric wake deflections with respect to the rotor's yaw angle. Examining the power of turbine array, [11] and [9] showed that only one direction of yaw misalignment resulted in a power increase of a two turbine array, while the exact opposite direction caused a power decrease. This finding has been confirmed by [3] experimentally using two model wind turbines. As those findings impact the applicability of the concept significantly, reasons for the asymmetry need to be understood. In this study, we show that a vertical velocity gradient has a direct effect on the wake's asymmetry during yaw misalignment using two model wind turbines in a wind tunnel study.

4. Please explain why a relatively unrealistic spacing between turbines (3D) is selected. In wind farms, turbine spacing usually falls in the range of 5D to 7D depending on terrain and flow conditions.

The experiments were performed at a wind tunnel of the University of Oldenburg, having a test section of 5 m length or  $\approx 8.6$  rotor diameters, whereas 5 m corresponds to the location of the collector. However, the spacing from the outlet/grid to the front turbine as well as the free stream configuration of the wind tunnel set limits the distance separating both turbines. In order to minimize wind tunnel effects due to the increasing shear layer of the free stream, the experiments were performed at a distance of x/D=3. We do agree that increasing distances would add valuable information, however, those were not performed due to the described wind tunnel limitations.

All of the following comments (5-9) address a lack of information that has been published in [3], where the same experimental setup was used apart from the sheared inflow profiles. Due to the limitations to 4 pages in length of the manuscript type 'Brief communication', we described only the most important aspects of the setup with the reference to [3] for more details. In general, we prefer to follow this principle due to the limitations and avoid describing details already published. However, we fully agree with the referee that some more very important aspects should be mentioned in the manuscript. In the following, a point-by-point response to the comments is given.

5. There is no information on how the turbine power is measured. Is it the electrical? Or the mechanical power extracted by the turbine form the wind?

The turbine power is  $P = T \cdot \omega$ , where  $\omega$  is the rotational speed and  $T = k \cdot I$  the torque based on the electric current I and the constant k = 79.9mN A-1 taken from the generator's specifications. The current I is measured by the voltage drop across a shunt resistor of 100 mΩ. Therewith, the power becomes  $P = \omega T = \omega k \frac{U_s}{0.1\Omega}$ .

This concept is described in [3] as shown by the screenshot in Figure 3 of this document, please refer to comment number 6 for the suggested update of the manuscript.

Data acquisition and turbine control were realized by a National Instruments NI-9074 cRIO real time controller equipped with modules for stepper motor control (NI-9512), analog input (NI-9215), analog output (NI-9264) and digital input/output (NI-9401) in combination with in-house built LabView software. The power of the model turbines,  $P = \omega T$ , is based on the generator's torque T, which is proportional to the electric current I according to the generator's specifications. I is obtained by measuring the voltage drop US across a shunt-resistor of 0.1  $\Omega$ . Therewith, the power becomes  $P := \omega k \frac{U_S}{0.1 \Omega}$ , where  $k = 79.9 \text{ mN A}^{-1}$  is the proportionality constant relating the generator's electric current to its torque.

Figure 3. Schreenshot taken from [3], description of power measurements.

**6. Please provide more information on about the wind tunnel (e.g. wind-tunnel type, test section size, and blockage ratio).**

We agree that this information is of importance and needs to be mentioned to a larger extent. The manuscript describes an experiments using the same setup is in a previous study [3], apart from the vertical velocity profiles. In [3], more detailed information about the setup are giving, which is shown in Figure 4 of this document.

**III. Experimental Setup**

Both turbines were placed in the wind tunnel of the University of Oldenburg with an outlet of 1 m x 0.8 m (width x height) and an open test section of 5 m length, displaced in streamwise direction as sketched in Figure 5. The distance x is variable, in this study we investigate the case x/D = 3. The outlet of the wind tunnel was equipped with an active grid as described by Weitemeier et al.17 The grid was used passively in open configuration with a blockage of nearly 4.8%, which resulted in a turbulence intensity of approx. 3% at hub height and  $u \approx 8 \text{ m s}^{-1}$ . The front turbine T1 was placed on a stepper motor driven turning table that allows a variation of the yaw angle. The wind speed  $u_2$ , which was the input wind speed for the load control of T2 as described in section B, was measured by a Prandtl tube 0.35 m in front of the downstream

aFurther characterizations showed that the maximal power coefficient achievable increases with the prevailing wind speed. Most likely, this is caused by mechanical losses, whose impact becomes less significant with increasing velocity.

4 of 8

American Institute of Aeronautics and Astronautics

Figure 4. Screenshot taken from [3], describing the setup.

Therefore, some aspect already described there were purposely not included in the current manuscript in order to keep the paper brief. A suggested update of Section 2 is given below:

p.2, ll.2 ff.:

The experiments were performed at a wind tunnel of the University of Oldenburg, with an open test section of  $1 \text{ m} \times 0.8 \text{ m} \times 5$ , m [w × h × 1]. Two model wind turbines as described by [12] were used in streamwise displacement. The turbines were separated by 3D, with D = 0.58 m being the rotor diameter. The upstream turbine is placed on a turning table allowing a yaw misalignment, while the where a positive yaw angle is a counter-clockwise rotation of the rotor observed from above. The downstream turbine utilizes a partial load control and therewith adapts to the changing inflow conditions. Power measurement are based on the rotational speed and the torque, being proportional to the electric current of the generator. Further details about the setup and power measurements are described by Schottler et al. (2016) [3]. In order to isolate ...

7. I suggest the authors to also test the performance of the turbines under uniform inflow conditions as a reference case. This can strengthen the authors' arguments. Moreover, Profile 2 down not have a good quality. It has a positive slope at

lower heights and a fairly negative slope a higher heights. A profile with a clearly negative slope (in contrast to profile 1) is more constructive.

The study [3] describes a very similar setup with the same grid installed, but all flaps being *open*, e.g. aligned with the main flow direction. Please refer to Figure 4 of this document for the exact passage. The results for the upstream and downstream turbine's power under uniform inflow conditions are discussed in this study. Figure 5 of this document shows a screenshot with the upstream and downstream turbine's power along with their sum. Here, also an asymmetry in  $P_2(\gamma_1)$ and  $P_{tot}(\gamma_1)$  is observed. The power of the upstream turbine  $P_1(\gamma_1)$  is shown to be close to symmetric. The three different sets show three measurements, showing the reproducibility of the results.

**Figure 5.**  $P_1$  and  $P_2$  (a) and  $P_{tot}$  (b) over  $\gamma_1$  during uniform inflow conditions, taken from [3].

8. Figure 2: Please add the variation of the power with the yaw angle for the upstream turbine. This helps readers to easier realize how yawing the upwind turbine reduces its own power and increases the power of the downwind one.

Figure 6 of this document shows Figure 2 of the manuscript with the power of the upstream turbine added to the plots. In our opinion, the plots appear a bit crowded now with three graphs overlapping. We suggest to normalize all graphs to the maximum value of  $P_{tot}$ , as done in Figure 7 of this document.

**Figure 6.** Mean values of  $P_1$ ,  $P_2$  and  $P_{tot}$  over  $\gamma_1$  for profile 1 (left) and profile 2 (right).

---

## Author Comment (AC2) · 29 Apr 2017

Authors' response to Referee #2, M. Paul van der Laan of DTU Wind Energy:

Dear Mr van der Laan, we, the authors, are very thankful for the detailed and constructive comments and greatly appreciate the willingness to review our manuscript. Especially, we would like to thank you for performing the numeric simulations shown in the comments. Please find our responses below. In this document, the original comments are shown in **bold** with the respective answers below. Excerpts of the manuscript are shown in *italic writing*, whereas additions are written in blue and deleted parts in .
Please note that the format of citations in manuscript excerpts might be changed.
Thank you very much for your efforts,

Jannik Schottler on behalf of all authors
* * *
Major comments:

1. **Where are the profiles from Figure 1 measured with respect to the wind turbine positions and how do they develop from the first to the second wind turbine and further downstream (without the wind turbines present in the tunnel). My concern is that if the wind profiles are far from equilibrium, it could influence the wake deflection significantly.**
   Thank you very much for the constructive concern. The hot wire array of the 13 sensors displaced vertically was installed at the upstream rotor's position, 1 m downstream of the inlet to the test section, before the turbine was installed. This is stated in p.2 ll.9-11 in the manuscript:

   *The downstream position of the hot wire array was 1 m from of the wind tunnel outlet, in agreement with the upstream turbine's rotor, which was installed after characterizing the inflow.*

   We suggest to formulate this more clearly in the revised manuscript as done below:

   *For both settings of the grid, data were recorded for 120 s at a sampling frequency of 2 kHz. The  array was installed 1 m *

 _downstream of the grid at the position of_ the upstream turbine's rotor, which was installed after characterizing the inflow.

We believe that stating the inflows are 'not in equilibrium' means that they will evolve further / change when moving downstream in the test section, even without any turbine installed. If that is what is meant, we fully agree with this concern and appreciate the constructive critic.

To create a boundary layer in a wind tunnel for experimental studies, often very long test sections ($>10\,\mathrm{m}$) are used to let a boundary layer develop due to inserted surface roughness elements, examples include Chamorro et al. (2009) [1] or Bastankhah and Porté-Agel (2016) [2]. Additionally, the cross sectional area is often adjusted for a zero pressure gradient. The work of Cekli and van de Water (2010) [3] gives a thorough overview and summarizes the problem precisely as quoted in Figure 1 of this document.

devices. A quite successful way to initiate a fat boundary layer with passive elements is through the "spires" described by Irwin (1981). These spires must be adapted to the desired flow profile.

Passive methods to simulate an atmospheric boundary layer in wind tunnels are still widely used in laboratories. Their main drawback is that usually a long test section is necessary to install all the vortex generators, roughness elements, etc. According to Simiu and Scanlan (1986), simulations done with the help of passive devices are not expected to result in favorable flow properties in short tunnels; however, a long test section wind tunnel may not be always available.

Several attempts have been reported to simulate an atmospheric boundary layer with active devices. Teunissen used an array of jets in a combination of barriers and roughness elements (Teunissen 1975). He could achieve

**Figure 1.** Screenshot taken from [3].

In our experimental setup, we are limited by the extension of the test section. However, the focus is not to create a realistic boundary layer profile, but to create inverse profiles by the usage of an active grid (used passively here). We do agree that in an ideal case both experimental capabilities, a long test section and therewith rather stable boundary layer as well as the possibility to inverse a profile, need to be combined. Achieving this experimentally is rather difficult and beyond our

experimental possibilities, which are limited by the test section length. However, using an active grid passively offers a great flexibility to purposely tune inflow gradients in shorter test sections. This is further described in [3]. It is important to notice that our work does not aim to create two realistic but inverted boundary layer profiles. It focuses on inverting an extreme shear profile, sacrificing certainty about the downstream development of the profiles.

We are aware of those limitations and therefore characterize the inflow profile at the exact same position as the upstream turbine in order to grasp the most appropriate inflow characteristics.

Depending on the downstream development, an influence on the wake deflection is possible. However, we do believe that the influence should be similar in both cases, profile 1 and profile 2. Unfortunately, we cannot prove this by measurement data. In order to minimize the possibility of wind tunnel effects to impair the findings significantly, we believe that it is a strength of the present study that between both tests cases, profile 1 and profile 2, all other aspects were kept the same, isolating the effect of the difference in inflow. Nevertheless, due to experimental limitations, it is hardly possible to fully distinguish the contribution of all parameters of the inflow, including turbulence parameters, all three velocity components, downstream development etc.

2. **What is the turbulence intensity and/or how do the turbulence profiles look like that correspond to profile 1 and 2 from Figure 1?**

   The profiles of the turbulence intensities $TI = \sigma_u / \overline{u}$ corresponding to Figure 1 of the manuscript are shown in Figure 2 of this document. As expected, the turbulence intensities increase where the flaps of the grid were not aligned with the main flow direction, e.g. lower velocities correspond to higher turbulence intensities. At the respective opposite side, where the flaps of the grid were in alignment with the main flow direction, the turbulence intensities are rather low, approximately 2-4 %.

   Due to the briefness of the manuscript, we suggest to leave Figure 1 of the manuscript as it is and restrain it the mean values of $u(z, t)$.

[Figure]

**Figure 2.** Turbulence intensities TI over height $z$ for the respective mean values shown in Figure 1 of the manuscript.

We want to thank the Reviewer for performing numeric simulations of a comparable scenario. We do agree, that it needs a combination of numeric, experimental (and field studies) to fully understand complex phenomena such as wake effects of/on wind turbines. In previous works by Gebraad et al. [4] and Fleming at al. [5], SOWFA[1] simulations were performed using a very similar setup of two aligned wind turbines, examining the power during a yaw misalignment of the upstream turbine. Here, large eddy simulations (LES) are linked to the aeroelastic tool FAST [8]. The SOFWA tool has been validated for example for an offshore wind farm by Chruchfield et al. (2012) [9]. Further studies include [10].

As in the simulations performed by the Referee, two NREL 5MW reference turbines were used, the distance separating both turbines was 7 rotor diameters, being notably larger than in the manuscript. At an inflow of $u = 8\,\mathrm{m\,s^{-1}}$, the vertical wind shear was $1.46\,\mathrm{m\,s^{-1}}$ across the rotor, corresponding to a natural boundary layer. For further details about the simulations, please refer to Fleming et al. (2014) [5]. For more details on SOFWA, see Figure 3 of this document.

Amongst others, the powers of both turbines were examined by Gebraad
* * *
[1]Simulator for Off/Onshore Wind Farm Applications, for further details, please see [6] or [7].

[Figure]

**Figure 3.** Screenshot taken from [11].

et al. in [4] for different angles of yaw misalignment of the upstream turbine, $\gamma_1$. Figure 4 of this document shows the results, taken from [4]. The

[Figure]

**Figure 4.** Screenshot taken from [4], Fig. 2, showing the power of an upstream turbine (blue), a downstream turbine (green, distance: 7D) and the total power of both (red) over the yaw angle of the upstream turbine.

power of the upstream turbine shows nearly symmetric variations with $\gamma_1$. The power of the downstream turbine, $P_2$ and the sum of both, $P_{tot}$, show distinct asymmetries. The minimum of $P_2$ is clearly shifted towards negative angles, resulting in an asymmetric total power. $P_{tot}$ is maximal at $\gamma_1 = 25°$, resulting in a power gain ($\approx 6\%$) as compared to $\gamma_1 = 0°$. Further, the opposite direction of yaw misalignment $\gamma_1 = -25°$ shows a power decrease compared to $\gamma_1 = 0°$.

Those principle shapes are in agreement with our experimental results presented in the manuscript as well as the results shown in [12]. To further show, Figure 5 of this document shows the normalized data taken from Gebraad et al. (2014) [4] and our experimental results for better comparison. The numerical data were received from P. Gebraad as a result of personal communications on this matter. It should be noted that the vertical wind

[Figure]

**Figure 5.** Comparison of the experimental results (left: profile 1, right: profile 2) and numerical results based on the data of [4]. All graphs are normalized to their maximum value.

shears are of *opposite* direction in the left plot of Figure 5 and of the *same* direction in the right.

Comparing our experimental results to the simulations of the Referee and the simulations from literature shown in Figures 4 and 5 reveal multiple aspects listed below:

- The simulations performed by Gebraad et al. show distinct asymmetries, although both turbines were (in the simulation environment)

aligned with the main flow direction *without* lateral offset. As shown by the reviewer, a lateral offsets could possibly cause asymmetries. However, this should not mean in turn that the asymmetries indicate a lateral offset of the turbines. This is shown by the simulation results in Figure 4. of this document.

The setup is sensitive to boundary conditions, but the turbines were aligned to our best possibilities.

- Comparing the simulations performed by the referee and the results shown in Figure 4, differences become apparent regarding the asymmetry of $P_2(\gamma_1)$ and $P_{tot}(\gamma_1)$. Although the same NREL 5 MW reference turbines were used, disparities seem to arise from other simulation set-ups, i.e. a different level of detail by using actuator line or actuator disc, boundary conditions of the setup, and/or turbine spacing. We believe those disparities show the need for further validation studies, either code-to-code validation or experimental work as done in our manuscript.

- Comparing simulations and experiment shown in Figure 5 of this document show similar trends. Looking at the left plot, both vertical wind shears are of *opposite* direction resulting in a very similar asymmetric shape but of reversed sign. On the right plot, both inflow shears were of the *same* direction, resulting in asymmetries where the minimum of $P_2$ is shifted to negative yaw angles and the total power $P_{tot}$ to positive yaw angles. Although full scale 5 MW turbines were simulated, having a larger spacing of 7D, the general shapes agree with the laboratory experiment using model turbines of much smaller scale and different spacing.
* * *
Minor comments:

1. **A few references include duplicated links.**
   This will be corrected in the updated manuscript.

2. **Page 1, lines 12-14: I am not able to find a discussion on asymmetries of wake deflection in Gebraad et al. (2016)**
   We appreciate pointing out this mistake, what was meant is the study of Gebraad et al. (2014) [4], not (2016). However, we want to be more precise in the updated manuscript. Vollmer et al. [13] investigated

wake deflections, while Gebraad et al. investigated the *power*, not the velocities, which was formulated somewhat unclear in the manuscript. We updated the manuscript accordingly as shown below. Please note that some other changes resulted from the comments of Reviewer#1.

*Lately, different concepts of active wake control are discussed throughout the research community. One promising concept is the wake deflection by intentional yaw misalignment of single wind turbines. The principle of deflecting the velocity deficit behind a wind turbine was observed in field measurements by [14], in wind tunnel experiments [15, 16] and in numerical simulations  [17, 4, 13] . Further, [18] and [19] applied the concept to wind farm control strategies using large-eddy simulation (LES) methods, showing a potential power increase in wind farm applications.*

*[13]  report on an asymmetric deflection of a turbine's wake with respect to its direction of yaw misalignment  in numeric studies. Similarly, [2] found that a wake moves upwards or downwards depending on the direction of a yaw misalignment using PIV measurements behind a small turbine model. This observation is explained by an interaction of the wake's rotation and a pair of counter-rotating vorticies formed in yawed conditions with the ground. [13] studied the influence of atmospheric stabilities on the wake deflection by yaw misalignment. The results showed that different stratifications indeed resulted in varying deflections of the wake behind the rotor of a numeric turbine model. More precisely, disparities between wake deflections due to yaw misalignments of +30° and −30° were significantly different considering different atmospheric stratifications and therewith different  shears. It is believed that a combination of a vertical inflow gradient, the wake's rotation and the wind veer cause asymmetric wake deflections with respect to the rotor's yaw angle. Examining the power of turbine array, [5] and [18] showed that only one direction of yaw misalignment resulted in a power increase of a two turbine array, while the exact opposite direction caused a power decrease. This finding has been confirmed by [12] experimentally using two model wind turbines. As those findings impact the applicability of the concept significantly, reasons for the asymmetry need to be understood.*

*In this study, we show that a vertical*  *wind shear has a direct effect on the wake's asymmetry during yaw misalignment using two model wind turbines in a wind tunnel study.*

3. **I would call vertical velocity gradient simply wind shear.**
   Thank you for suggesting this simpler formulation. In order to be precise about the direction of shear, we suggest to reformulate this to *vertical wind shear* in the updated version of the manuscript.

4. **How is your yaw angle defined?**
   Thank you very much for this hint. Some information about the setup were left out as the study [12] uses the same setup apart from the inflow variations. However, we absolutely agree that this should be mentioned in the manuscript besides the reference to [12]. We suggest to update the manuscript as done below. It should be noted that other changes in this paragraph result from the comments of the first referee.

   p.2 ll. 4 ff.:
   *Two model wind turbines as described by [20] were used in streamwise displacement. The turbines were separated by $3D$, with $D = 0.58\,\text{m}$ being the rotor diameter. The upstream turbine is placed on a turning table allowing a yaw misalignment,*  *where a positive yaw angle is a counter-clockwise rotation of the rotor observed from above. The downstream turbine utilizes a partial load control and therewith adapts to the changing inflow conditions. Power measurement are based on the rotational speed and the torque, being proportional to the electric current of the generator. Further details about the setup and power measurements are described by Schottler et al. (2016) [12].*

5. **Page 2, lines 14-16: I would add over the rotor area to be more precise: Using two inflows which feature a vertical velocity gradient in opposite direction over the rotor area allows an investigation of ....**

   We do agree that this formulation would add clarity. This will be done in the updated version of the manuscript as shown below:

   P.2, ll. 14-16:
   *Using two inflows which feature a vertical velocity gradient in opposite direction over the rotor area allows an investigation of the gradient's influence on the asymmetric power output of the two turbines with respect to the upstream turbine's yaw angle, $\gamma_1$.*

**References**

[1] Chamorro, L. P. and Porté-Agel, F., "A wind-tunnel investigation of wind-turbine wakes: boundary-layer turbulence effects," *Boundary-layer meteorology*, Vol. 132, No. 1, 2009, pp. 129–149.

[2] Bastankhah, M. and Porté-Agel, F., "Experimental and theoretical study of wind turbine wakes in yawed conditions," *Journal of Fluid Mechanics*, Vol. 806, No. 1, nov 2016, pp. 506–541.

[3] Cekli, H. E. and Water, W. V. D., "Tailoring turbulence with an active grid," *Experiments in Fluids*, Vol. 49, No. 2, 2010, pp. 409–416.

[4] Gebraad, P., Teeuwisse, F., Wingerden, J., Fleming, P., Ruben, S., Marden, J., and Pao, L., "Wind plant power optimization through yaw control using a parametric model for wake effects—a CFD simulation study," *Wind Energy*, 2014.

[5] Fleming, P., Gebraad, P. M., Lee, S., Wingerden, J.-W., Johnson, K., Churchfield, M., Michalakes, J., Spalart, P., and Moriarty, P., "Simulation comparison of wake mitigation control strategies for a two-turbine case," *Wind Energy*, 2014.

[6] Churchfield, M. and Lee, S., "NWTC Information Portal (SOWFA)," https://nwtc.nrel.gov/SOWFA, Accessed 17-April-2017.

[7] Fleming, P. a., Gebraad, P. M. O., Lee, S., van Wingerden, J. W., Johnson, K., Churchfield, M., Michalakes, J., Spalart, P., and Moriarty, P., "Evaluating techniques for redirecting turbine wakes using SOWFA," *Renewable Energy*, Vol. 70, No. June, 2014, pp. 211–218.

[8] Jonkman, J. M. and Buhl Jr, M. L., "FAST user's guide," *National Renewable Energy Laboratory, Golden, CO, Technical Report No. NREL/EL-500-38230*, 2005.

[9] Churchfield, M., Lee, S., Moriarty, P., Martinez, L., Leonardi, S., Vijayakumar, G., and Brasseur, J., "A Large-Eddy Simulations of Wind-Plant Aerodynamics," *50th AIAA Aerospace Sciences Meeting including the New Horizons Forum and Aerospace Exposition*, , No. January, 2012, pp. 1–19.

[10] Churchfield, M. J., Lee, S., Michalakes, J., and Moriarty, P. J., "A numerical study of the effects of atmospheric and wake turbulence on

wind turbine dynamics," *Journal of Turbulence*, Vol. 13, No. June 2015, 2012, pp. N14.

[11] Churchfield, M., Michalakes, J., Spalart, P., and Moriarty, P., "Evaluating techniques for redirecting turbine wake using SOWFA," 2013.

[12] Schottler, J., Hölling, A., Peinke, J., and Hölling, M., "Wind tunnel tests on controllable model wind turbines in yaw," *34th Wind Energy Symposium*, , No. January, 2016, pp. 1523.

[13] Vollmer, L., Steinfeld, G., Heinemann, D., and Kühn, M., "Estimating the wake deflection downstream of a wind turbine in different atmospheric stabilities: an LES study," *Wind Energy Science*, Vol. 1, No. 2, sep 2016, pp. 129–141.

[14] Trujillo, J.-J., Seifert, J. K., Würth, I., Schlipf, D., and Kühn, M., "Full field assessment of wind turbine near wake deviation in relation to yaw misalignment," *Wind Energy Science Discussions*, , No. January, 2016, pp. 1–17.

[15] Medici, D. and Alfredsson, P., "Measurements on a wind turbine wake: 3D effects and bluff body vortex shedding," *Wind Energy*, Vol. 9, No. 3, 2006, pp. 219–236.

[16] Krogstad, P.-Å. and Adaramola, M. S., "Performance and near wake measurements of a model horizontal axis wind turbine," *Wind Energy*, Vol. 15, No. 5, 2012, pp. 743–756.

[17] Jiménez, Á., Crespo, A., and Migoya, E., "Application of a LES technique to characterize the wake deflection of a wind turbine in yaw," *Wind energy*, Vol. 13, No. 6, 2010, pp. 559–572.

[18] Gebraad, P. M. O., Teeuwisse, F. W., van Wingerden, J. W., Fleming, P. A., Ruben, S. D., Marden, J. R., and Pao, L. Y., "Wind plant power optimization through yaw control using a parametric model for wake effects-a CFD simulation study," *Wind Energy*, Vol. 19, No. 1, jan 2014, pp. 95–114.

[19] Fleming, P. A., Ning, A., Gebraad, P. M. O., and Dykes, K., "Wind plant system engineering through optimization of layout and yaw control," *Wind Energy*, Vol. 19, No. 2, feb 2016, pp. 329–344.

[20] Schottler, J., Hölling, A., Peinke, J., and Hölling, M., "Design and implementation of a controllable model wind turbine for experimental studies," *Journal of Physics: Conference Series*, Vol. 753, sep 2016, pp. 072030.

---

## Referee Report (RR1)

**Review of revised version Brief Communication: On the influence of vertical wind shear on the combined power output of two model wind turbines in yaw by J. Schottler, et al.**

Reviewer: M. Paul van der Laan, DTU Wind Energy

June 9, 2017

The article presents wind tunnel measurements of two aligned model wind turbines. The influence of two different inflow profiles on the power output is investigated, as function of the yaw misalignment angle of the upstream wind turbine.

I would like to point out that there is a typo in my previous review. The introduced wind turbine offset in the simulations corresponds to only $\pm\text{atan}(1/(8\times 3) = \pm 2°$ (and not 7° as previously reported). This indicates that a very small misalignment angle can cause an asymmetry in the power vs yaw plots, as shown by Reynolds-averaged Navier-Stokes simulations performed in the first review.

The authors have correctly responded to all the comments; however, I do believe that more discussion of the author's response can be added to the article. I would suggest to add the following to the discussion:

- The downstream development of the inflow profiles has not been measured. A downstream development of the inflow profiles can have an impact on the wake deflection of the upstream wind turbine. This can also lead to asymmetries of the power of the downstream wind turbine with respect to yaw of the upstream wind turbine.

It is good that you refer to observations of numerical large eddy simulations (LES); however, one should be aware of following effects that can also lead to asymmetries of the wake deficit in LES:

1. In LES, the inflow wind direction is a distribution, which could have a mean wind direction that has a small offset at the wind turbine position.

2. The referenced LES articles also include wind veer.

You could consider to add these comments to the article if you find them relevant.

To summarize, I would recommend to accept the article if the authors add the suggested discussion about the downstream development of the measured inflow profiles.

---

## Referee Report (RR2)

2nd round of the review of the paper entitled

*"Brief Communication: On the influence of vertical wind shear on the combined power output of two model wind turbines in yaw"*

I would like to thank the authors for their efforts to address my and the other reviewer's comments. However, I still have a few concerns regarding the experimental setup employed in this work, namely:

- Based on the response of the authors, the upstream turbine operates under constant loading (constant $U_{FET}$). As a result, TSR decreases with the yaw angle as shown in Fig. 1 of the response letter. This means that comparison of the turbine power production at different yaw angles can be questionable as turbines operate at different (not necessarily optimum) TSRs. This may not affect asymmetry in the power output reported in the manuscript. However, the results are indeed more reliable if, first, the value of the load corresponding to the optimal TSR is found for each yaw angle.
- Based on Fig. 2 in the response letter, the value of the turbulence intensity of Profile 2 at the turbine hub height level is more than two times of the one for Profile 1. Due to this significant difference in the turbulence level between Profiles 1 & 2, the wake of the upstream turbine can have a totally different recovery rate depending on the incoming profile. This in turn affects the power production of the downstream turbine. One has to therefore compare the power output for Profiles 1 and 2 with caution.
- In response to the other reviewer, the authors acknowledged that the incoming boundary layers are not fully developed. I think it is useful if the authors mention this limitation in the manuscript with more quantitative information (e.g., variation of velocity in the streamwise direction without the presence of the turbines). This helps readers to bear this limitation in mind when they try to interpret the presented results.

Minor comments:

- Please update the caption of Fig. 2 in the manuscript, following the changes made in this figure.

Overall, I think the paper is well written and it is worth of publication. However, the above-mentioned limitations should be first addressed, or at least they have to be clearly stated in the manuscript for the sake of completeness.

---

## Author Response (AR2)

This document contains the responses to the reports of Referees #1 and #2 as well as a marked-up version of the manuscript showing the changes made to the previous version.

The authors want to thank the referees for their time of effort to review the revised version of the manuscript. This is greatly appreciated. Please find our responses below. The original comments are shown in **bold** with the respective answers below. Excerpts of the manuscript are written in *italic*, aspects added to the manuscript are written in blue.
Thank you very much,

Jannik Schottler on behalf of all authors
* * *
Responses to Referee #1:

1. **The authors have correctly responded to all the comments; however, I do believe that more discussion of the author's response can be added to the article. I would suggest to add the following to the discussion:**
   **The downstream development of the inflow profiles has not been measured. A downstream development of the inflow profiles can have an impact on the wake deflection of the upstream wind turbine. This can also lead to asymmetries of the power of the downstream wind turbine with respect to yaw of the upstream turbine.**

   Thank you very much, we strongly agree that parts of the discussion of the first round of reviews would improve the paper and should be added to the discussion in the manuscript. Stating that the downstream development of the flow was not investigated previously will be addressed in the revised version of the manuscript. Thus we add to the discussion section, p. 4:

   *[...]. Further, due to spatial limitations of the wind tunnel, the profiles shown in Fig. 1 are expected to be not fully developed. Therefore, their downstream development, which was not investigated in this study, might impact the wake deflections. This effect could not be isolated. [...]*

2. **It is good that you refer to observations of numerical large eddy simulations (LES); however, one should be aware of fol-**

lowing effects that can also lead to asymmetries of the wake deficit in LES:

(a) In LES, the inflow wind direction is a distribution, which could have a mean wind direction that has a small offset at the wind turbine position.

(b) The referenced LES articles also include wind veer. You could consider to add these comments to the article if you find them relevant.

Thank you for those comments. Regarding point (a), I think the fact that the yaw angle is a distribution and not fixed should be addressed. However, I believe this should not be limited to the LES simulations as neither [1] nor [2] give exact information about a distribution of the yaw angle. Also, it is of relevance when comparing full scale cases with experiments and probably an important topic on its own. We add to the end of the discussion, p. 4, ll 29 f:

*[...], such as detailed wake measurements during different inflow gradients and yaw errors. As the yaw angle is a distribution in full scale cases, future works should address this issue and its impact on active wake redirection strategies.*

Regarding point (b), the wind veer should be included in the discussion, as it is a relevant difference between the referenced LES simulations and the experiments. According to the comment we write (p. 4):

*[...] It should be noted that the LES simulations performed in [1] and [3] include a wind veer, which was not reproduced experimentally and should be kept in mind when comparing the numerical and experimental studies. [...]*
* * *
Responses to Referee #2:

1. **Based on the response of the authors, the upstream turbine operates under constant loading (constant $U_{FET}$). As a result, TSR decreases with the yaw angle as shown in Fig. 1 of the response letter. This means that comparison of the turbine**

**power production at different yaw angles can be questionable as turbines operate at different (not necessarily optimum) TSRs. This may not affect asymmetry in the power output reported in the manuscript. However, the results are indeed more reliable if, first, the value of the load corresponding to the optimal TSR is found for each yaw angle.**

Thank you for your comment. We agree that adding aspects of the peer-review discussion to the discussion of the manuscript would add quality to the paper. This should include a statement of the changing TSR due to yaw misalignment. The model turbine's controller can set a constant TSR by adapting the voltage $U_{FET}$, therewith the angle of attack at the blade is kept constant for $\gamma = 0°$ and idealized inflow conditions, with $\gamma$ being the yaw angle. During yaw misalignment, a blade section experiences permanent angle of attack changes due to induced velocities, which makes the controller principle of keeping the TSR constant somewhat questionable and the *optimal* TSR becomes a non-trivial parameter. For example, Krogstad & Adaramola [4] report of a decreased optimal TSR (optimal = TSR($c_{P,max}$)) during yaw misalignment. This is likely to be dependent on the airfoil / blade design used. We add this aspect to the manuscript (p. 4):

*[...]. It should also be noted that the upstream turbine's tip speed ratio (TSR) is not constant for varying angles $\gamma_1$. As shown by [4], the TSR maximizing the power is subject to change with the yaw angle. Therefore, the load control utilized by the downstream turbine was not used for the upstream turbine, which was operated at constant electrical load for both profiles. However, as the upstream turbine's TSR is symmetric with respect to $\gamma_1$, this is not expected to affect the asymmetries observed in this work.*

2. **Based on Fig. 2 in the response letter, the value of the turbulence intensity of Profile 2 at the turbine hub height level is more than two times of the one for Profile 1. Due to this significant difference in the turbulence level between Profiles 1 & 2, the wake of the upstream turbine can have a totally different recovery rate depending on the incoming profile. This in turn affects the power production of the downstream turbine. One has to therefore compare the power output for Profiles 1 and 2 with caution.**

Thank you for pointing this out. We do recognize that statistical properties of the flow beyond its mean values (over height) are of importance regarding wake effects, including wake recovery. Those properties include the turbulence intensity. In order to ideally isolate an effect, all other relevant properties should be equal. However, using the active grid to create both profiles experimentally sets limits to what flow properties can be controlled simultaneously. Indeed, due to the different TIs, one has to be careful when comparing absolute power values. However, we do not expect the asymmetries of the powers for the respective profiles to be affected. We fully agree that this should be stated in the discussion section, we therefore add to the manuscript, p.4:

*[...]. Next, the inflow profiles vary regarding their turbulence intensity. This is expected to impact the wake recovery [5], but not the asymmetries in power reported.*

3. **In response to the other reviewer, the authors acknowledged that the incoming boundary layers are not fully developed. I think it is useful if the authors mention this limitation in the manuscript with more quantitative information (e.g., variation of velocity in the streamwise direction without the presence of the turbines). This helps readers to bear this limitation in mind when they try to interpret the presented results.**

We agree with this comment and will add this point to the discussion section of the manuscript, p.4:

*[...]. Further, due to spatial limitations of the wind tunnel, the profiles shown in Fig. 1 are expected to be not fully developed. Therefore, their downstream development, which was not investigated in this study, might impact the wake deflections. This effect could not be isolated. [...]*

4. **Please update the caption of Fig. 2 in the manuscript, following the changes made in this figure.**

Thank you for the hint, this will be corrected.

[revised manuscript text omitted]